# A Role for iNOS in Erastin Mediated Reduction of P-Glycoprotein Transport Activity

**DOI:** 10.3390/cancers16091733

**Published:** 2024-04-29

**Authors:** Shalyn M. Brown, Birandra K. Sinha, Ronald E. Cannon

**Affiliations:** Laboratory of Mechanistic Toxicology, Division of Translational Toxicology, National Institute of Environmental Health Sciences, Research Triangle Park, Durham, NC 27709, USA; shalyn.brown@nih.gov (S.M.B.); birandra.sinha@nih.gov (B.K.S.)

**Keywords:** erastin, ferroptosis, iNOS, P-glycoprotein, blood–brain barrier

## Abstract

**Simple Summary:**

Erastin is a small molecule that was discovered in a screen to selectively kill cancer cells. Mechanistic studies of the cancer cells killed by erastin determined their death to be dependent on iron, increased levels of lipids and oxidative stress. This type of cell death was called “ferroptosis”. We hypothesized that even low levels of erastin would lead to changes in signaling pathways in the cell. To test this, we exploited an ex vivo blood–brain barrier transport assay and confirmed our findings in vitro in human tumor cells. We determined that nanomolar levels of erastin could induce specific biological changes in the activity of an important transporter (pump) at the BBB and in tumor cells. We showed that erastin rapidly inhibited the activity of the Multidrug-Resistant Protein (MDR1, P-glycoprotein). Noteworthy, P-glycoprotein is responsible for most of the chemo-resistance in cancer therapies.

**Abstract:**

The blood–brain barrier is composed of both a physical barrier and an enzymatic barrier. Tight junction (TJ) proteins expressed between endothelial cells of brain capillaries provide the physical barrier to paracellular movement of ions and molecules to the brain, while luminal-facing efflux transporters enzymatically restrict the entry of blood-borne molecules from entering the brain. The expression and activity of ATP Binding Cassette transporters or “ABC” transporters in endothelial cells of the BBB and in human tumor cells are dynamically regulated by numerous signaling pathways. P-glycoprotein (P-gp), (ABCB1), is arguably the most studied transporter of the BBB, and in human cell lines. P-glycoprotein transport activity is rapidly inhibited by signaling pathways that call for the rapid production of nitric oxide (NO) from the inducible nitric oxide synthase enzyme, iNOS. This study investigated how nano-molar levels of the selective chemotherapeutic erastin affect the activity or expression of P-glycoprotein transporter in brain capillaries and in human tumor cell lines. We chose erastin because it signals to iNOS for NO production at low concentrations. Furthermore, erastin inhibits the cellular uptake of cystine through the X_C_^−^ cystine/glutamate antiporter. Since previous reports indicate that NO production from iNOS can rapidly inhibit P-gp activity in tumor cells, we wondered if induction of iNOS by erastin could also rapidly reduce P-glycoprotein transport activity in brain endothelial cells and in human tumor cell lines. We show here that low concentrations of erastin (1 nM) can induce iNOS, inhibit the activity of P-glycoprotein, and reduce the intracellular uptake of cystine via the Xc- cystine/glutamate antiporter. Consistent with reduced P-glycoprotein activity in rat brain capillary endothelial cells, we show that human tumor cell lines exposed to erastin become more sensitive to cytotoxic substrates of P-glycoprotein.

## 1. Introduction

This study investigated the immediate effect of low concentrations (0.001–1.0 µM) of erastin on the transporters of the blood–brain barrier (BBB) in rats. The blood-brain barrier (BBB) resides in the vast array of microvessels of the brain. Structurally, the BBB is arranged as a tri-laminate, consisting of luminal-facing endothelial cells with an outward layering of pericytes and astrocytic feet [1]. Tight junctions between the luminal-facing endothelial cells provide a formidable physical barrier, while luminal-bound transporters discretely allow entry of needed nutrients and chemicals that serve the biological needs of the brain. Unwanted and potential harmful blood-born molecules are defended against by a host of luminal-bound ATP-binding cassette (ABC) transporters that selectively restrict xenobiotics, drugs, and harmful metabolites from entering the brain. ABC transporters hydrolyze ATP to ADP + Pi to derive sufficient energy to efflux harmful substrates back into the circulating blood. Arguably, the most important and most studied ABC transporter of the BBB is P-glycoprotein (ABCB1, MDR1) [2]. P-glycoprotein is a 170 kD monomeric protein highly expressed on the luminal endothelial plasma membrane that is dynamically regulated at the activity and expression level through a vast array of signaling pathways. It is also highly expressed in cancer cells and is the primary contributor to chemotherapeutic drug resistance in the clinic [3,4]. It has an expansive and diverse array of substrates that include prescription drugs, xenobiotics, and metabolites [5].

The BBB responds to a myriad of stress responses through the activation of biological signaling pathways. These pathways regulate BBB function by altering the expression and/or activity of ABC transporters at the barrier [6]. Since previous work has shown that oxidative stress has the capacity to modulate transport activity at the BBB [7], we investigated whether erastin could also modulate transport activity at the BBB. To accomplish this, we exposed freshly isolated rat brain capillaries ex vivo to low nanomolar concentrations of erastin and measured P-glycoprotein transport activity using a steady state confocal microscopy-based assay. Furthermore, we extended our findings to humans by exposing cancer cell lines in vitro to erastin and measuring the cytotoxic effect of a known cytotoxic P-glycoprotein substrate.

The chemotherapeutic, erastin (2-[1-[4-[2-(4-Chlorophenoxy)acetyl]-1-piperazinyl]ethyl]-3-(2-ethoxyphenyl)-4(3*H*)-quinazolinone), was identified in a selective screen of small molecules capable of killing cancer cells that overexpress small-T antigen and oncogenic RAS [8]. Cancer cells exposed to micromolar levels of erastin die from a non-apoptotic death that is associated with membrane lipid peroxidation, increased oxidative stress, reduced antioxidative capacity, and high levels of iron. This erastin-induced, iron-dependent form of programmed cell death was named “ferroptosis” [9]. Studies show that erastin can induce ferroptosis through multiple pathways that include but are not limited to perturbations of the system X_C_^–^ (glutamate /cystine antiporter) [10], the mitochondria-bound voltage-dependent anion channel (VDAC) [11], and the tumor suppressor p53 gene [12]. Mechanistic studies on ferroptosis found that cancer cells exposed to micromolar levels of erastin were depleted of the important antioxidant, glutathione (GSH) [13]. The depletion of intracellular GSH was found to result from erastin’s ability to inhibit the plasma membrane-bound glutamate/cystine antiporter, system X_C_^−^. The X_C_^−^ antiporter functions to import extracellular cystine in exchange for an abundant intracellular glutamate [13]. Upon entry, intracellular cystine is rapidly reduced to cysteine and utilized in the synthesis and maintenance of intracellular GSH levels to help protect against oxidative stress. We chose erastin for our studies because it had been previously shown to induce iNOS, presumably through its interaction with the Xc- antiporter. We hypothesized that NO production from erastin mediated iNOS induction could affect the activity and/or expression of P-glycoprotein at the BBB. 

## 2. Methods

### 2.1. Animals

All animal experiments presented in this work were approved by the Animal Care and Use Committee at the National Institute of Environmental Health Sciences in accordance with the National Institutes of Health. The data derived from these experiments are presented in compliance within the guidelines of the Animal Research Reporting In Vivo Experiments (ARRIVE). All male and female Sprague Dawley rats aged 12–50 weeks were purchased from Envigo and housed in humidity- and temperature-controlled rooms with a 12 h light/dark cycle. All were provided with access to both food and water ad libitum. Rats were killed by the approved method of CO_2_ inhalation and decapitation. 

### 2.2. Materials

Ficoll, Bovine Serum Albumin (BSA), sodium pyruvate, L-Cysteine, and D-(+)-glucose, were all purchased from Sigma-Aldrich (Saint Louis, MO, USA). Thirty-micrometer pluriStrainer filters were purchased from pluriSelect, and 300 µm nylon mesh was purchased from Spectrum Labs. Erastin was purchased from Sigma-Aldrich. The P-glycoprotein fluorescent substrate N-ɛ(4-nitrobenzofurazan-7-yl)-D-Lys8 cyclosporin A (NBD-CSA) was provided as a gift from Dr. Bjoern Bauer from the University of Kentucky. The BCRP fluorescent substrate BODIPY^®^ FL prazosin was purchased from Invitrogen (Waltham, MA, USA), and the MRP2 fluorescent substrate sulforhodamine 101 free acid (Texas Red, St. Louis, MO 68178, USA) was purchased from Sigma-Aldrich, St. Louis, MO, USA. PSC833 (valspodar), the P-glycoprotein-specific inhibitor, was purchased from Tocris Bioscience. Two-well 1.5 chamber slides were purchased from Thermo Fisher Scientific (Waltham, MA, USA). The antioxidant N-acetylcystine (NAC) was gifted by Alex Merrick. Ten-well Invitrogen 4–12% Bis-Tri NuPAGE Gels NP0321, Novex Nitrocellulose membranes LC2001, the XCell *SureLock*^®^ Mini-Cell and the XCell II™ Blot Module were all purchased from Thermo Fisher Scientific. Odyssey^®^ Blocking Buffer was obtained from Li-Cor Biosciences (Lincoln, NE, USA). Immobilon-FL membranes were purchased from Sigma-Aldrich. Mouse monoclonal P-glycoprotein antibody C219 was purchased from Covance (Princeton, NJ, USA). Secondary goat anti-mouse IgG Alex Fluor 488 antibodies were purchased from Invitrogen. Mouse monoclonal β-actin antibody A1978 was purchased from Sigma-Aldrich. The iNOS antibody was purchased from Invitrogen. The iNOS inhibitor 1400 W was obtained from Abcam (Cambridge, UK), and the pan-NOS inhibitor L-NAME was obtained from Cayman (Ann Arbor, MI, USA). Thermo Scientific MES, 0.5 M buffer solution, pH 6.0 was purchased from Thermo Fisher Scientific. The reducing agent tris(2-carbosyethyl)phosphine hydrochloride, the cystine analog seleno-L-cystine, the fluorescent molecule fluorescein O,O’-diacrylate, and the system X_C_^−^ inhibitor sulfasalazine were all purchased from Sigma-Aldrich. 

### 2.3. Cell Culture 

Authenticated human ovarian tumor cells, OVCAR-8 and NCI/ADR-RES, were obtained from the National Cancer Institute (Bethesda, MD, USA) and were grown in Phenol Red-free RPMI 1640 media supplemented with 10% fetal bovine serum and antibiotics. Tumor cells were routinely used for 20–25 passages, after which the cells were discarded and a new cell culture was started from the frozen stock.

### 2.4. Capillary Isolation

Brain capillaries were isolated from male and female Sprague Dawley rats. Using an established protocol [14], rats were killed with CO_2_ inhalation and subsequently decapitated. Rat brains were extracted from the skull and placed in a cold PBS buffer solution containing KCl 2.7 mM, KH_2_PO_4_ 1.5 mM, NaCl 136.9 mM, Na_2_HPO_4_ 4.3 mM, CaCl_2_ 0.7 mM, MgCl_2_ 0.5 mM, augmented with D-glucose 5.0 mM, and sodium pyruvate 1.0 mM at pH 7.4. Dissection of the rat brains involved removal of meninges, midbrain, olfactory bulbs, choroid plexus, blood vessels, and cortical white matter. Remaining grey matter brain tissue was homogenized in cold PBS buffer solution using a tissue grinder followed by ten up and down strokes in a Dounce homogenizer. The resulting homogenate was centrifuged in an aliquot of 30% Ficoll for 20 min at 5800× *g* at 4 °C, separating the lipid fraction from the brain parenchyma. The capillary pellet was resuspended in 1% BSA in PBS, passed through a 300 μm nylon mesh filter, and then passed through a series of 30 μm pluriSelect cell strainers. Capillaries were washed off the filters with PBS to remove them and resuspended, followed by a second centrifugation of 350× *g* for 10 min at 4 °C. Following final centrifugation, the supernatant was removed, and capillaries were used immediately for transport or protein studies.

### 2.5. Transport Assay

Following capillary isolation, the pellet was resuspended in a cold PBS buffer solution and plated into 2-well chamber slides. Capillaries could then be treated with various antagonists, agonists, or inhibitors. For dosing experiments, serial dilutions of erastin were prepared immediately before use in PBS buffer solution using 10.0 mM erastin stock solution made in DMSO. Solutions containing the fluorescent substrate specific for P-glycoprotein (NBD-CSA) and appropriate dosing of erastin were then added to respective chamber slides at volumes of 2.0 mL. Time course studies were staggered such that each slide could be viewed consecutively on a confocal microscope at 15 min/slide. For studies involving BCRP and MRP2, fluorescent substrates BODIPY^®^ FL prazosin and sulforhodamine 101 free acid (Texas Red) were used for the transporters, respectively. Transport activity was measured as a function of steady-state luminal fluorescence. In the P-gp studies, background fluorescence was measured using the P-gp-specific inhibitor, PSC833 10 μM (valspodar), and subtracted from raw treatment values to obtain specific luminal fluorescence as a result of inhibition or activation of the transporter. 

### 2.6. Confocal Microscopy

Capillaries were imaged using the Zeiss LSM 710 multiphoton confocal microscope and the Zeiss LSM 880 confocal microscope through a 40× water-immersion objective with a numeric aperture of 1.5. A 488 nm laser line for both NBD-CSA and BODIPY^®^ FL prazosin and a 543 nm laser line for Texas Red were used. The resulting images were saved to a data storage drive, which were then analyzed and quantified using ImageJ software. This published method was described in detail by Chan and Cannon in 2017 [14]. 

### 2.7. Western Blotting

The capillary isolation protocol was outlined previously. Following isolation, capillary pellets were treated with 1.0 nM erastin for 3 h, then centrifuged at 350× *g* for 10 min at 4 °C and placed in a −80 °C freezer until further use. Thawed samples were separated into cytosolic, membrane, and nuclear fractions using lysis buffer with a protease inhibitor. Membrane and cytosolic protein fractions were mixed with loading dye, sample reducing buffer, and water and loaded onto a 4–12% Bis-Tris NuPAGE gel. Samples were subsequently electrophoresed using the XCell *SureLock*^®^ Mini-Cell and transferred to an Immobilon-FL membrane using the XCell II™ Blot Module according to the manufacturer’s instructions. Following transfer, blocking buffer was added to the membrane for 40 min. The membrane was rinsed with 0.1% Tween in PBS, treated with mouse or rabbit monoclonal primary antibodies, and allowed to hybridize overnight. After overnight hybridization, the membrane was then rinsed with 0.1% Tween in PBS and stained with goat anti-mouse or goat anti-rabbit secondary antibodies for 1 h. The membrane was rinsed in three 40-min intervals with 0.1% Tween in PBS and imaged using an Odyssey infrared imaging system from Li-Cor Biosciences. Images were analyzed, and values were normalized to the β-actin loading control. 

### 2.8. Cystine Uptake Assay via X_C_^−^ Antiporter 

The capillary isolation protocol was outlined previously, and the cystine uptake assay via X_C_^−^ was adapted for brain capillaries. Once capillary isolation was completed, capillary pellets were transferred to a microfuge tube and washed with pre-warmed, cystine-free, serum-free media. Capillaries were centrifuged at 350× *g* for 10 min at 4 °C. The supernatant was removed, and capillaries were resuspended in 200 µL of pre-warmed cystine-free, serum-free media and divided into microfuge tubes for a variety of treatments. Capillaries were centrifuged again at 350× *g* for 10 min at 4 °C, and the supernatant was removed. A total of 200 µL of pre-warmed cystine-free, serum-free media was then added to each microfuge tube in addition to pretreatments of erastin and sulfasalazine, both known inhibitors of the X_C_^−^. Erastin was added at a dose of 1.0 nM, while sulfasalazine was added at a dose of 500 µM. Tubes were shielded from light and allowed to sit for 15–20 min. Following pretreatment, tubes were incubated at 37 °C for 5 min and then centrifuged at 350× *g* for 10 min at 4 °C. The supernatant was then removed, and 200 µL of pre-warmed cystine analog solution (200 µM selenocysteine) was added to each microfuge tube, except for the controls. To the respective tubes, 1.0 nM erastin and 500 µM sulfasalazine were added. To the controls, 200 µL of cystine-free, serum-free media was added. The tubes were incubated at 37 °C for 30 min. The supernatant was removed, and capillaries were rinsed three times with 200 µL of ice-cold PBS solution. Supernatant was removed and 60 µL of 100% methanol was added to each tube. A working solution was prepared immediately before use that consisted of 100 mM MES buffer (pH 6.0), 10.0 µM fluorescein O,O’-diacrylate, and 200 µM tris92-carboxyethyl)phosphine hydrochloride. A total of 240 µL of working solution was added to each microfuge tube, mixed by pipetting, and incubated a second time for 30 min at 37 °C. Following incubation, capillaries were centrifuged at 20,800× *g* for 5 min at 4 °C. The final supernatant was transferred to new microfuge tubes, then plated into a Greiner Fluotrac flat-bottomed 96-well black plate. Fluorescence was read using a BioTek Synergy microplate reader. Fluorescence intensity was measured at Ex/Em = 490/535 and samples were read in triplicate. 

### 2.9. Cytotoxicity Assay

The cytotoxicity studies were carried out with both a cell growth inhibition assay and Trypan Blue. Briefly, 50,000–75,000 cells/well were seeded onto a 6-well plate (in triplicate) and allowed to attach for 18 h. Various concentrations of drugs (adriamycin or combinations of adriamycin, erastin or NOS inhibitors) were added to cells (NCI/ADR-RES) in fresh complete media (2 mL) and incubated for 72 h. When used, ER and NOS inhibitors were preincubated with cells for 2 h before the addition of Adriamycin. DMSO (0.01–0.1%) was included as the vehicle control when used. Following trypsinization, surviving cells were collected and counted in a cell counter (Beckman, Brea, CA, USA) or 15 μL of cell mixtures were combined with 15 µL of trypan blue and counted in a T20 automatic cell counter (Bio-Rad, Hercules, CA, USA). 

### 2.10. Statistical Analysis

Data were graphed and analyzed using the GraphPad Prism (v 9.2.0) software and expressed as mean ± SEM. Statistical analyses and determination of significance were performed by one-way ANOVA and the Tukey multiple comparison test. Significance was defined as *p* < 0.05. Values deemed significant were so when compared to control, unless otherwise specified. 

## 3. Results

### 3.1. Erastin Decreases in P-Glycoprotein Transport Are Dose- and Time-Dependent 

Using a previously established confocal microscopy-based assay [14], we measured the activity of the ABC transporter, P-glycoprotein at the BBB of male and female rats. This was accomplished by measuring the luminal accumulation of the P-glycoprotein-specific fluorescent substrate NBD-CSA at steady state in freshly isolated rat brain capillaries (Figure 1A). Non-specific (background fluorescence) was removed by subtracting the residual nonspecific fluorescence of capillaries treated with 10.0 μM PSC833, a highly selective P-glycoprotein inhibitor (Figure 1B). Using this method, we determined the effect of increasing concentrations of erastin on P-glycoprotein transport activity at the BBB by exposing freshly isolated rat brain capillaries from male and female rats to 0.001–1.0 µM erastin for 3 h and measuring P-glycoprotein transport activity (Figure 1C–E). In males, P-gp transport activity was lower than the untreated controls across all concentrations of erastin. However, only 0.001 µM and 1.0 µM erastin concentrations significantly lowered P-glycoprotein transport activity. In females, only 0.001 µM erastin was found to significantly lower P-glycoprotein transport activity. Next, we investigated the kinetics of the erastin-dependent changes in P-glycoprotein transport by treating capillaries with 0.001 µM erastin for 1–4 h and measuring P-gp transport hourly (Figure 1F,G). We chose 0.001 µM erastin because it produced the most significant reduction in P-gp transport activity across both sexes. We found that capillaries exposed to 0.001 µM erastin significantly lowered P-gp transport activity in 3 h in males and 1 h in females.

Having established that the erastin-induced decreases in P-glycoprotein transport activity are dose- and time-dependent, we wondered if this response was shared among other ABC transporters of the BBB. Using our transport assay (Figure 2A,B), we exposed male and female rat brain capillaries to 0.001 µM erastin for 3 h and measured BCRP and MRP2 transport activity using their respective fluorescent substrates (Bodipy-prazosin and Texas Red).

In male capillaries exposed to 0.001 µM erastin for 3 h, we observed significant increases in BCRP and MRP2 transport activity. This contrasted to females, where we observed significant changes in MRP2 transport but no changes in BCRP transport. We can conclude two important facts from these results. First, the effect of erastin on ABC transporters at the BBB is unique to each. Second, erastin’s associated decreases in P-glycoprotein transport activity were not due to physical damage that altered capillary permeability or leakage. If this were the case, a decrease in luminal fluorescence would have occurred, and the transport activity for all the transporters would be reduced. 

### 3.2. Erastin’s Effects on P-Glycoprotein Transport Activity Are Reversible

Next, we performed a transport reversibility assay to explore the mechanistic nature of the erastin-induced decreases in P-glycoprotein transport. To perform this assay, we exposed male and female rat brain capillaries to 0.001 µM erastin for 3 h, measuring transport activity hourly. After 3 h, we remove erastin and continue to measure transport activity hourly for 3 additional hours. We posit that if the observed erastin-induced decreases in P-glycoprotein transport activity are due to protein turnover or changes in expression, the transport activity would not rapidly revert to control levels upon chemical removal. However, if the changes are due to chemical perturbation of signal transduction pathways, independent of expression changes in P-gp, then we would expect a rapid return to basal (control) levels upon erastin removal. As shown in Figure 3A,B, P-glycoprotein transport activity rapidly reverted to control levels within 1 h after the removal of erastin for both sexes. 

The results from our reversibility assay suggested that erastin-mediated decreases in P-glycoprotein transport activity at the BBB were not due to changes in transporter protein levels. To confirm this, we first performed qualitative Western blots to access P-glycoprotein levels in male and female rat brain capillaries that were vehicle control treated or 0.001 µM erastin treated for 3 h. No qualitative differences in P-glycoprotein levels were detected in females between erastin-treated or untreated control capillary membrane lysates (Figure 4A and Appendix A); however, P-glycoprotein levels in males appeared slightly reduced in the treated samples (Figure 4A). To examine this male response a bit closer, we performed quantitative westerns using three independent treatments and normalized the loading to beta-actin (Figure 4B and Appendix A). The band densities were measured and graphed in Figure 4C and Appendix A. Taken together, these results indicate that no significant changes in P-glycoprotein result from 3-h erastin (0.001 μM) treatment in male or female rat brain capillaries.

### 3.3. Erastin-Reduced Intracellular Cystine via System X_C_^−^


In cancer cells, erastin at concentrations of 10.0 µM or higher is used to inhibit the X_C_^−^ cystine/glutamate antiporter, leading to a depletion of intracellular cystine and glutathione levels [10]. To date, there are no reports of erastin eliciting biological effects below 1.0 µM, and none have been shown to affect transporters of the BBB. To determine if nM concentrations of erastin are affecting the X_C_^−^ cystine/glutamate antiporter of the BBB endothelium, we adapted a fluorescence-based assay to measure changes in cystine uptake [15]. An illustration of this assay is shown in Figure 5A. The assay determines the activity of the X_C_^−^ cystine/glutamate antiporter by quantifying the intercellular levels of the imported cystine analogue, selenocystine. This is accomplished by reacting the intracellular contents, after uptake, with tris(2-carboxyethyl)phosphine (TCEP) and fluorescein *O*,*O*′-diacrylate (FOdA) under weakly acidic conditions (pH 6.0). Fluorescence intensities measured at 535 nm (Fl. units) of FOdA after reacting with selenocystine and TCEP are directly proportional to the imported levels of selenocystine via the X_C_^−^ antiporter. We used this assay to determine if exposures to nM concentrations of erastin affected cystine uptake via the X_C_^−^ antiporter. As previously described above, we measured selenocystine uptake in male rat brain capillaries exposed to vehicle control or to 1.0 nM erastin for 30 min.

As shown in Figure 5A, capillaries exposed to erastin 1.0 nM were significantly inhibited (~10,000 Fl. units) by selenocystine uptake through the system X_C_^−^ antiporter when compared to vehicle control-treated capillaries (~20,000 Fl. units). These data also indicate that 1.0 nM erastin was not as potent an inhibitor of the system X_C_^−^ antiporter as sulfasalazine at 500 µM (100 Fl. units). 

Having established that exposure to 1.0 nM erastin reduced the levels of cystine through the system X_C_^−^ antiporter, we hypothesized that lower cystine levels might be involved in the erastin-induced reductions in P-glycoprotein transport activity at the BBB. To test this, we measured P-glycoprotein transport activity in male and female rat brain capillaries treated for three hours with either vehicle control, 1.0 nM erastin, 300 µM cysteine or both 300 µM cysteine and 1.0 nM erastin. As shown in Figure 5B,C, P-glycoprotein transport activity in both sexes was significantly reduced by 3 h of exposure to 1.0 nM erastin. However, in the cotreatments, cysteine completely abolished erastin’s repressive effect on P-glycoprotein transport activity in both sexes.

To further explore the role of cysteine and GSH, we duplicated the previous experiment using N-acetylcysteine (NAC) for cysteine. NAC is a prodrug like cysteine that is known to replenish intracellular GSH levels [16]. We found that co-treating with NAC (200 µM) also completely abolished erastin’s repressive effect on P-glycoprotein transport activity (Figure 5D,E).

### 3.4. NOS Inhibitors Blocked the Effects of Erastin on P-Glycoprotein Transport

Work on the BBB has identified numerous signaling pathways that rapidly repress P-glycoprotein transport activity independent of protein expression changes [6,17,18]. Upon close inspection, most shared iNOS as a required component. Additionally, it has been recently shown in tumor cells that donor compounds that release NO can rapidly inhibit the ATPase activity of P-glycoprotein [19,20]. Recently, human breast cancer cells exposed to erastin were found to have increased levels of Inos [21]. Given these findings, we wondered if increases in iNOS were associated with erastin’s induced repression of P-glycoprotein in rat brain capillaries. Knowing that endothelial cells could potentially express three forms of NOS, we selected two inhibitors of NOS activity to use in inhibitor studies: L-NAME (100 μM), a non-selective pan-NOS inhibitor, and 1400 W (10 μM), a highly selective iNOS inhibitor. We isolated rat brain capillaries from both sexes and measured P-glycoprotein transport activity after exposing them for 3 h to a vehicle control, NOS inhibitors alone, or cotreatments with 0.001 µM erastin. As shown in Figure 6A males and B females, erastin reduced P-glycoprotein transport activity in three hours compared to vehicle control. Interestingly, both the pan inhibitor and the highly selective iNOS inhibitor abolished the erastin-repressive effects on P-glycoprotein transport, suggesting that iNOS is involved. To further investigate this, we used Western blots to measure the levels of iNOS in untreated male rat brain capillaries compared with those treated with 1.0 nM erastin (Figure 6C and Appendix A). Given that the effects of the NOS inhibitors were identical in both males and females, we chose to only observe the protein levels of iNOS in male rats. 

Our quantitative Western blots show significant increases in iNOS protein levels in the erastin-treated compared to untreated samples, further suggesting that iNOS was induced by low levels of erastin. Taken together, our inhibitor and Western blot data are consistent with the notion that iNOS activity is required for the erastin induced repression of P-glycoprotein at the BBB. 

### 3.5. Human Tumor Cells Treated with Erastin Increased Cell Killing by Adriamycin

To determine if erastin affects P-glycoprotein transport activity in human cells, similar to our findings at the BBB, we performed cell survival assays on two ovarian human-derived cell lines, OVCAR-8 and NCI/ADR-RES. The OVCAR-8 cell line is devoid of P-gp expression, while the NCI/ADR-RES cells express high levels of P-gp. To assess the effect of erastin on P-glycoprotein activity in these cells, we measured the percent survival of both cell lines grown in media with increasing concentrations of the cytotoxic P-glycoprotein substrate, adriamycin (10^−9^–10^−4^ M). To evaluate the effect of erastin and the role of iNOS, we co-dosed the cell lines with erastin and 1400 W. The data (Figure 7) show that 10^−5^ M adriamycin alone allows for 75% cell survival.

Co-dosing the cells with increasing concentrations of adriamycin with 100.0 nM erastin significantly increased the killing of 10^−5^ M adriamycin to 40% cell survival. Significant is the observation that inhibiting iNOS with 1400 W under identical treatment conditions markedly increased cell survival from 40% to 60%. These data are consistent with our findings in rat brain capillaries, showing that erastin reduces P-glycoprotein transport activity in human cells. P-glycoprotein transport activity leads to an increase in adriamycin cytotoxicity in human cells. Also, analogous to our findings in rat brain capillaries, inhibiting iNOS with 1400 W in human cells abolished the erastin-induced repression of P-glycoprotein transport. 

## 4. Discussion

Erastin was originally discovered in a screen of small molecules exhibiting the ability to selectively kill cancer cell lines expressing *SV40* small T antigen and oncogenic *ras* while leaving normal cells unharmed [8]. The unique erastin-induced cell death of cancer cells was called ferroptosis and was determined to involve membrane lipid peroxidation, glutathione depletion, and high levels of iron [9]. Further studies have shown that erastin kills cancer cells in a three-pronged attack: (1) inhibiting the uptake of cystine through the system X_C_^−^ antiporter [10], (2) disrupting mitochondria function and releasing oxidants from the voltage-dependent anion-selective channel (*VDAC*) [11], and (3) modulating the expression and activity of the tumor suppressor gene, TP53 [12]. The cumulative effect of these attacks weakens cancer cells’ oxidative defenses by depleting GSH while simultaneously inducing higher levels of oxidants. These events occur in cancer cells that contain high levels of ionic iron (Fe^+2^) that freely participate in Fenton reactions to produce oxidized membrane lipids [22]. 

A mechanistic understanding of erastin-induced ferroptosis is important beyond the therapeutic needs of killing cancer cells in the clinic. Numerous pathophysiological disease states closely mimic aspects of ferroptosis [23,24]. These include, but are not limited to, blood diseases, central nervous system (CNS) diseases, ischemia and reperfusion injuries, and kidney injuries. A greater understanding of molecular mechanisms that induce, inhibit, and regulate ferroptosis will help to mitigate the deleterious effects of many diseases and cancers.

In this study, we investigated the rapid effects of low nM concentrations of erastin on the BBB of rats and the tumor cell lines of humans. Specifically, using freshly isolated rat brain capillaries, we measured the effects of nM concentrations of erastin on P-glycoprotein transport activity and expression at the BBB. We found that erastin caused a dose- and time-dependent reduction in P-glycoprotein transport activity in both male and female rats. The maximal reductions in transport occurred in 3 h in capillaries that were exposed to 1.0 nM erastin. One interesting aspect of the erastin reduction in P-glycoprotein transport is that males were not significantly reduced until 3 h of exposure to 1.0 nM erastin, while P-glycoprotein transport in females was significantly reduced at 1 h of exposure and sustained thereafter. The reason for this early and sustained response to erastin in females is unknown. Additionally, it can be noted that in both male and female rats, P-glycoprotein transport activity appears to slightly increase following the 3-h mark, suggesting that when treated with nM concentrations of erastin, P-gp transport activity could reverse without removal of erastin. These observed sex differences of erastin on P-glycoprotein activity could be related to the intrinsic differences in the basal and/or inducible NO levels in males compared to females. Multiple studies have shown that systemic NO is higher in females than males [25,26]. The observed biphasic response of P-glycoprotein as erastin concentrations approach micromolar levels may also be related to the multiple biological downstream effectors that are known targets of erastin at higher concentrations. These are known to produce cellular stressors related to oxidative stress, mitochondrial dysfunction and dysregulation of important stress response genes like P53 [9,12]. Similar stressors have been associated with signaling pathways that increase P-glycoprotein transport activity over longer periods of time [6].

Mechanistically, our experiments indicate that the erastin-induced reductions in P-glycoprotein at the BBB are not due to physical damage or loss of barrier integrity. Our work also shows that erastin is transporter-specific. For example, under identical exposure conditions that rapidly reduce P-glycoprotein activity in males, we observed erastin-induced increases in BCRP and MRP2 transport activities. However, in females, only MRP2 transport activity was increased by 1.0 nM erastin, indicating that the effects of erastin on the transporters of BBB are transporter- and sex-specific. Our experiments also show that erastin-dependent reductions in P-glycoprotein transport activity are rapidly reversed upon erastin removal. This rapid reversal back to basal levels suggests the involvement of signaling pathways that affect transport activity without changing protein levels of expression. Our Western blots, measuring protein levels, confirmed that erastin treatments did not alter P-glycoprotein expression levels. These data taken together suggest that erastin-induced reductions in P-glycoprotein transport activity were not due to protein degradation. The data are more consistent with the notion that erastin is modulating important signaling pathways that regulate transport activity at the BBB.

To understand the effect of nM concentrations of erastin in BBB endothelial cells, we used a novel fluorescent-based assay that measured the uptake of selenocystine through the system X_C_^−^ antiporter [15]. To our knowledge, this is the first time it has been used on brain microvessels. All previous reports exposed cells to high µM concentrations of erastin to cause inhibition of cystine uptake. Using freshly isolated rat brain capillaries, we found that low nM amounts of erastin rapidly inhibited cystine uptake. We logically reasoned that lower cysteine levels could contribute to GSH depletion. Furthermore, we wondered if reduced GSH levels were also responsible for the rapid decreases in P-glycoprotein transport activity we observed in our rat brain capillaries. To address this, we co-treated capillaries with 1.0 nM erastin plus NAC or cysteine, two known precursors to GSH. We found that co-treating with either alone blocked the repressive effects of erastin on P-glycoprotein transport activity. These experiments suggested that oxidative stress in the absence of sufficient levels of GSH could activate signaling pathways that altered the pump activities of important transporters at the BBB (e.g., P-glycoprotein). 

In endothelial cells, oxidative stress can lead to increases in NO production. This is especially true during high and rapid states of cytokine release from inflammatory conditions such as rheumatoid arthritis (RA) [27] or chronic brain disorders such as multiple sclerosis (MS) [28]. This associative link between oxidative stress and NO (NOS activity) extends to previously described signaling pathways that regulate P-glycoprotein in rat brain capillaries [7]. We found that human cancer cells treated with NO donor compounds become more sensitive to killing by P-glycoprotein substrates [29]. These findings are consistent with the notion that NO is involved (required) in the inhibition of P-glycoprotein transport activity. Lastly, it has been recently reported that human cancer cells exposed to erastin exhibit increases in iNOS activity and expression [21]. Our Western blots measuring iNOS protein levels clearly show that iNOS is rapidly induced by nM levels of erastin. Furthermore, we found either the pan-NOS inhibitor (L-NAME) or the selective iNOS inhibitor (1400 W) blocked the erastin-induced inhibition of P-glycoprotein transport activity. These observations are consistent with previous studies involving iNOS-containing signaling pathways that rapidly inhibit P-glycoprotein transport activity.

To summarize our work, we have proposed a model that integrates our findings and helps to understand how erastin rapidly reduces P-glycoprotein activity at the BBB. As shown in Figure 8, our findings indicate that nM levels of erastin reduce the uptake of cystine through the system X_C_^−^ antiporter. This has been previously reported to occur in other cell types, but at concentrations that are 10–100 times higher [30]. Inhibition of P-glycoprotein (ABCB1) by erastin has also been previously reported in human ovarian cancer cells [31]. The suggested mechanism in these studies involved substrate competition. Their work also used erastin exposures of 48 h at 1.0–30.0 µM. We have not evaluated erastin at high µM concentrations in brain capillaries, but our work using low nM concentrations of erastin does not support a competitive substrate model of inhibition. Also, antioxidants and NOS inhibitors blocked the effects of erastin on P-glycoprotein in our experiments, eliminating a competitive model and supporting the notion that erastin is activating a signaling pathway to inhibit P-glycoprotein transport activity. 

Two other possible erastin-mediated perturbations we have not examined are TP53 and/or mitochondrial VDAC function. We cannot eliminate the possibility that either or both may be involved in our observed effects of erastin. However, we found that exogenous NAC, or cysteine, completely blocks the repressive effects of erastin on P-glycoprotein transport, suggesting that the loss of cystine uptake is the primary and causal event. Loss of intracellular cystine and subsequent decreases in GSH are known to lead to oxidative stress. We show that in rat brain capillaries exposed to erastin, there is an increase in NO production via iNOS. Based on previous and present work, we believe that NO inhibits the ATPase activity of P-glycoprotein [19]. Further work will be needed to confirm the identity of the nitrosylation sites on P-glycoprotein. However, we have previously identified putative sites within the ATPase domains that could be involved [29]. This work is important because of the essential function the BBB plays in protecting the brain. Clinically, it suggests that erastin could rapidly lower P-glycoprotein activity at the BBB to facilitate the entry of drugs to reach targets within the brain/CNS. This work also suggests that erastin could be useful as an adjunct in cancer therapies by lowering chemoresistance (MDR), rendering cancer cells more sensitive to certain chemotherapeutic drugs. This could be clinically important given that the majority of chemotherapeutics are P-glycoprotein substrates. 

## Figures and Tables

**Figure 1 cancers-16-01733-f001:**
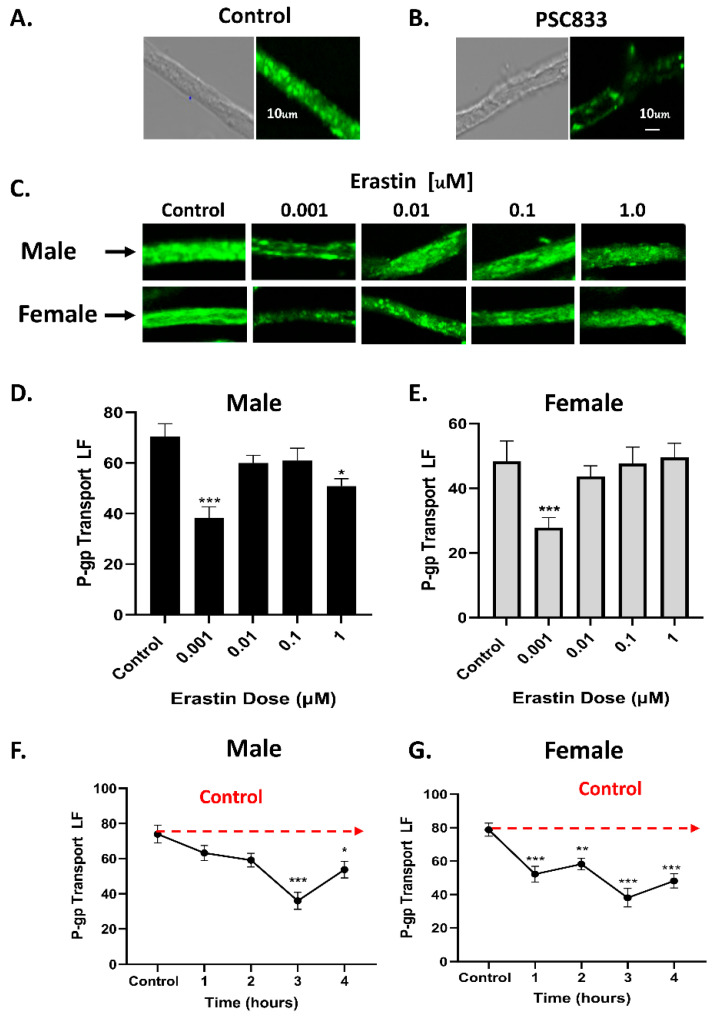
Changes in P-glycoprotein (P-gp) transport activity in brain capillaries (*N* = 15–20 per data point) from six rats treated with erastin. (**A**) Representative confocal fluorescent and DIC images of brain capillaries after a 60-min incubation with 2.0 μM NBD-CSA; note the high luminal fluorescence in the control capillary and (**B**) decreased luminal fluorescence in capillaries exposed to 10.0 μM PSC833 (Scale bars, 10 μm). Representative confocal images of the luminal fluorescence (LF) from (**C**) male and female brain capillaries associated with P-glycoprotein (P-gp) transport activity at increasing doses of erastin in brain capillaries of SD rats. Bar graphs of P-glycoprotein transport activity determined by LF in (**D**) male and (**E**) female capillaries. Graphs of LF measuring P-glycoprotein (P-gp) transport activity in male (**F**) and female (**G**) capillaries (capillary *N* = 15–20 per experimental data point) exposed to 0.001 μM erastin over time. Dotted red line denotes multiple comparison. Significance is as compared to control unless otherwise specified: * *p* < 0.05, ** *p* < 0.01, *** *p* < 0.001.

**Figure 2 cancers-16-01733-f002:**
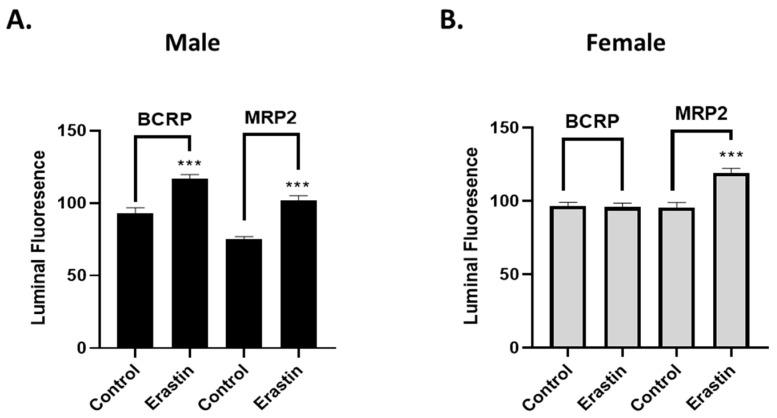
Breast Cancer Resistance Protein (BCRP) and Multidrug Resistance Protein 2 (MRP2) transport activities in brain capillaries (*N* = 15–20) from six rats treated with erastin. (**A**) Males and (**B**) females are graphs of BCRP and MRP2 transport activity in brain capillaries of SD rats treated with 0.001 μM erastin for 3 h. SE and significance were determined by one-way ANOVA and Tukey multiple comparison. Significance is as compared to control unless otherwise specified: *** *p* < 0.001.

**Figure 3 cancers-16-01733-f003:**
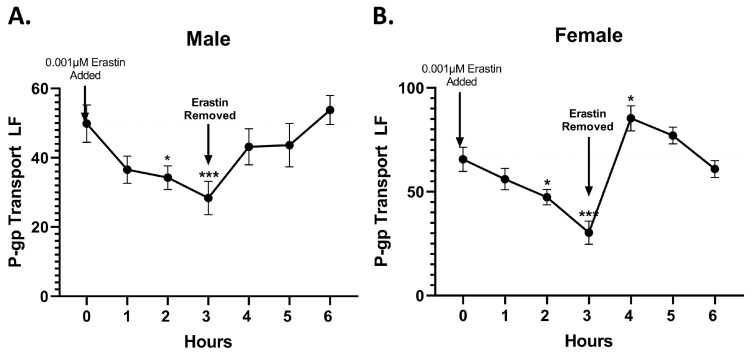
Transport reversibility following erastin removal. Capillaries treated with erastin (0.001 μM) for 3 h then removed. (**A**) Male and (**B**) female brain capillaries are graphs showing measured P-glycoprotein (P-gp) transport activities in brain capillaries (*N* = 15–20 per data point) from six rats before and after erastin removal. SE and significance were determined by one-way ANOVA and Tukey multiple comparison. Significance is as compared to control unless otherwise specified: * *p* < 0.05, *** *p* < 0.001.

**Figure 4 cancers-16-01733-f004:**
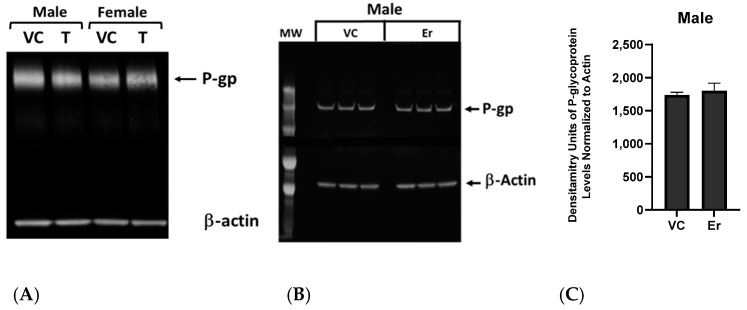
Determining P-glycoprotein (P-gp) levels after erastin (0.001 μM) treatment. (**A**) Qualitative Western blots determined P-glycoprotein (P-gp) protein levels in erastin (Er), and vehicle control (VC) treated male and female SD rats (Animal N# = 6 pooled/well) brain capillary membrane lysates. (**B**) Quantitative P-glycoprotein Western blot of male rat brain capillary lysates performed in triplicate and measured by normalization to actin (6 rat brains/well). (**C**) Graph of P-glycoprotein band density units in males comparing erastin-treated (Er) to vehicle control (VC). Band density levels are found in Appendix A.

**Figure 5 cancers-16-01733-f005:**
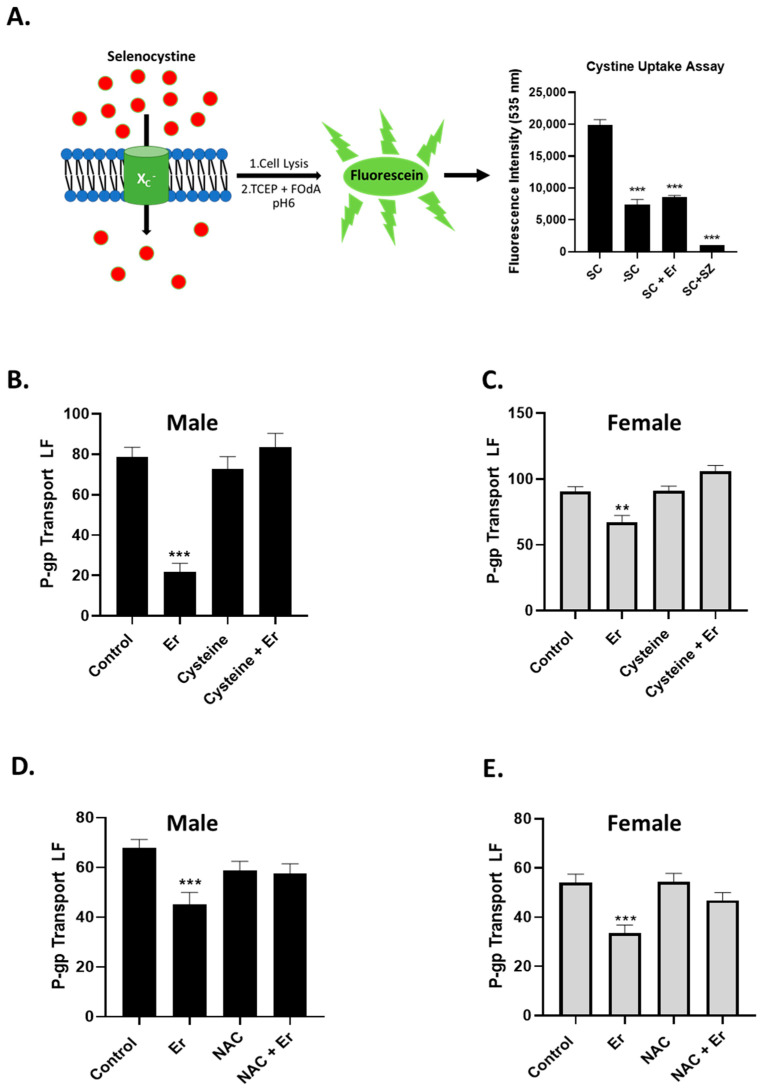
Measuring erastin’s effect on cystine uptake and co-dosed with antioxidants, cysteine, or NAC. (**A**) (**Left**) Illustration of the assay used to measure X_C_^−^ antiporter activity. (**Right**), Graph of measured fluorescence produced from selenocystine uptake through the X_C_^−^ antiporter. Measured fluorescence of untreated rat brain capillaries with selenocystine (SC) or without selenocystine (-SC). SC + Er, selenocystine with erastin (0.001 μM). SC + SZ, selenocystine with 500.0 μM sulfasalazine an X_C_^−^ inhibitor. (**B**) Male and (**C**) Female rat brain capillaries (*N* = 15–20) isolated from 6 rats per experiment and exposed to 0.001 μm erastin (Er), cystine alone (Cystine) or cystine with erastin (Cystine + Er). (**D**) Male and (**E**) female rat brain capillaries (*N* = 15–20) exposed to 0.001 μm erastin (Er), NAC alone or NAC with erastin (NAC + Er). SE and significance were determined by one-way ANOVA and Tukey multiple comparison. Significance is as compared to controls: ** *p* < 0.01, *** *p* < 0.001.

**Figure 6 cancers-16-01733-f006:**
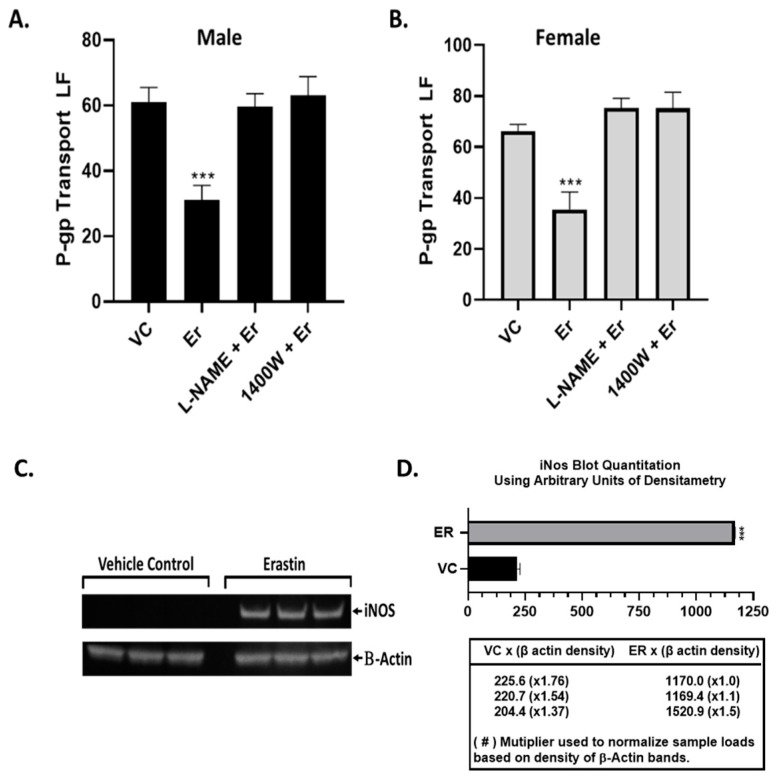
iNOS induction is required for erastin’s decrease in P-glycoprotein transport activity. Measurements of P-glycoprotein transport activity in (**A**) male and (**B**) female rat brain capillaries (*N* = 15–20) from 6 rats were treated with vehicle control (VC), erastin (Er), erastin with 100 μM of pan-NOS inhibitor L-NAME (L-NAME + Er), and erastin with 10 μM of the selective iNOS inhibitor 1400 W (1400 W + Er). (**C**) Western blot of vehicle and 0.001 μM erastin-treated male rat brain capillaries (pooled from 6 rats per well) and (**D**) graph of band density of iNOS normalized to loading control β-actin levels. SE and significance were determined by one-way ANOVA and Tukey multiple comparison. Significance is as compared to controls: *** *p* < 0.001.

**Figure 7 cancers-16-01733-f007:**
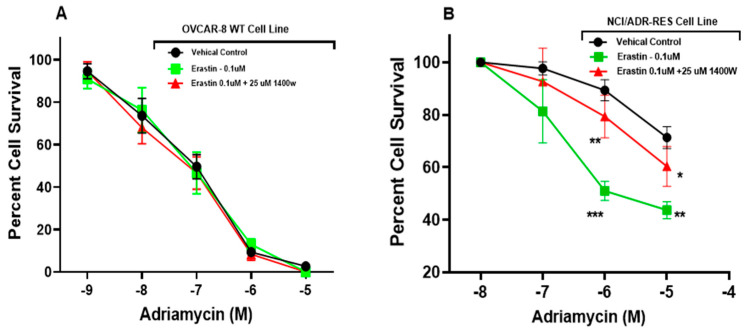
Erastin increases the cytotoxicity of P-gp substrate in human cells in vitro that express P-glycoprotein. Graph representing (**A**) the percent of OVCAR-8 (non-expressing P-glycoprotein) and (**B**) NCI/ADR-RES (P-glycoprotein)-expressing cells surviving 72 h following adriamycin (0.01–10.0 μM) exposures with or without erastin (100 nM) or with erastin (100 nM) co-dosed with the iNOS inhibitor, 1400 W. Values represent three separate experiments carried out in triplicate. SE and significance were determined by one-way ANOVA and Tukey multiple comparison. Significance is as compared to control: * *p* < 0.05, ** *p* < 0.01, *** *p* < 0.001.

**Figure 8 cancers-16-01733-f008:**
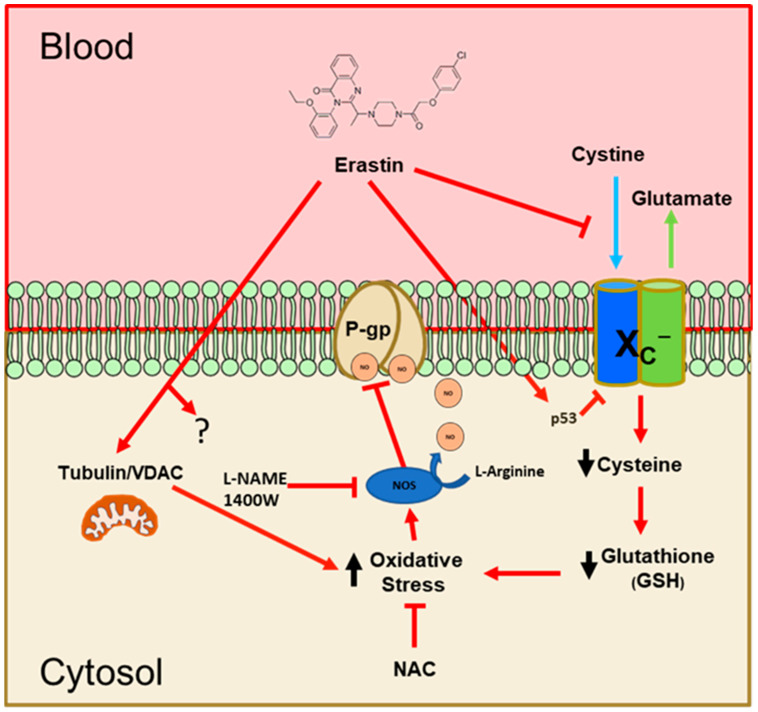
Proposed model of how erastin reduces P-gp transport. Erastin blocks transport of cystine into the capillary via system X_C_^−^, decreasing intracellular cystine concentration and leading to a decrease in precursors for glutathione (GSH) production within the capillary. Decreases in GSH lead to increased oxidative stress, which indirectly leads to increases in iNOS activity that nitrosylate residues within P-glycoprotein. Although not a focus of this work, erastin also affects ion channels (VDAC) of the mitochondria and dysregulates TP53 expression and activity, which influences the expression of key genes important in both death and survival. Exogenous addition of the GSH precursors (NAC/cystine) mitigates the inhibitory effects of erastin on P-glycoprotein transport activity.

## Data Availability

The data presented in this study are available in this article (and Appendix A).

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
