# Peer review of "A Role for iNOS in Erastin Mediated Reduction of P-Glycoprotein Transport Activity"

_cancers, 2024, doi:10.3390/cancers16091733_

Round 1

Reviewer 1 Report (Previous Reviewer 1)

Comments and Suggestions for Authors

If the authors would like to keep the fig 5 I have no objections.It was a minor point.

Author Response

We appreciate your time and effort to make our manuscript better. We have tried to correct   and address all your concerns. 

This manuscript is a resubmission of an earlier submission. The following is a list of the peer review reports and author responses from that submission.

Round 1

Reviewer 1 Report

Comments and Suggestions for Authors

This is an interesting paper reporting some new findings about the regulation of Pgp at the Blood Brain Barrier, however the paper is written in a somewhat confusing assembly of different results including also non-BBB cells. Thus the title does not represent thefull content of the paper. Also in the abstract the cystine glutamate antiporter results are only mentioned in a small clause. In the body of the text this topic covers almost 2 pages. If it is so important as the authors state in their discussion this should be mentioned in the title and the abstract. Mainly however the 2 effects should be correlated in more detail to each other. More experiments are necessary to prove this. The same is true for the effect of erastin on Pgp in other human cells not belonging to the BBB. If I take this serious, then the observed effects are not specific for the BBB. This should be mentioned in the abstract.

Objections given in detail:

1) The authors used concentrations of erastin between 1nM and 1uM claiming that erastin inhibits Pgp transport activity. They observe, that for the capillaries obtained from the male animals 1nM has the highest effect, higher doses are less effective except 1uM. No explanation is given for this strange effect. In the corresponding capillaries from females only 1nM is effective, higher doses have no effect. This is rather unconventional and should be explained or better should be investigated on a molecular basis. Moreover since 1nM is the only concentration that has been shown to be effective, it is not possible to state, that the effect is concentration dependent. This should be shown in a range around 1 nM. May be determination of the NO level helps to understand the effects.
2) What is the explanation that erastin has no effect on BCRP in the capillaries from females?
3) Is the BCRP and the MRP2 effect dose dependent?
4) The time dependence in fig 3 before erastin removal is different from the one shown in fig 1.
5) Fig 5 must not be shown
6) Why is the effect of erastin on the Pgp activity shown in fig 5B much bigger compared to the results shown before or in fig 5D? This seems to be not consistent with the arrow bars
7) In the experiments shown in fig7 0.1uM erastin was used. What about the lower concentration e.g 1nM?
8) Why are higher elastin concentrations not used at least to compare the effects with literature data (see discussion)
9) The ATPase activity of Pgp should be tested as mentioned in the discussion.

Reviewer 2 Report

Comments and Suggestions for Authors

General Comments

Although it is important to investigate inhibitory effects of P-glycoprotein transport activity by chemotherapeutic agents and the underlying mechanism, there are several major concerns that need to be addressed for the study, please see the specific comments below.

Specific Comments

1.     Where is iNOS located? Since the wall of brain capillaries is named the blood-brain barrier, which consists of endothelial cells, pericytes and astrocyte foot processes, which types of cells express iNOS, only endothelial cells? If so, any evidence?

2.     Fig. 1 is a little blurring. Please upload a clear one. What is the percentage of LF in Fig. 1B (background) to that in Fig. 1A?

3.     It is kind of puzzling that the higher dose erastin induces less effect on P-gp activity but longer time treatment with the same low dose of erastin induces more effect on P-gp activity (Figs.1 C-F), any explanation? It is better to show the confocal images of capillaries with P-gp luminal fluorescence under the treatment with various doses of erastin, corresponding to Figs. 1C,D.

4.     In Figs. 1E,F, the P-gp transport LF was measured on the same sample at different times?

5.     In the caption of Fig. 1B, it said that 5.0 uM PSC833 was used, but in P5, l7 lines below Results, it said that 10.mM PSC833 was used, which one is correct?

6.     The last sentence in P5 is not complete.

7.     In Figs.1,2, how many capillaries were used for each case, from how many rats?

8.     In Fig. 3, it was said that capillaries were from 6 rats? 6 rats each for male and female, or 3 rats for male and 3 for female? How many capillaries were used in each case?

9.     In Fig. 4, it was said SD rats (N=6) were used, male or female, or both?

10.  In Fig. 6C,D, another set of experiments with a higher erastin dose, such as 1 uM, which does not reduce P-gp activity in the capillaries of female rats (Fig. 1D) should be done to show if this higher dose of erastin increases iNOS. If it does, then the statement that iNOS activity is required for the nM erastin induced repression of P-gp at the BBB is not quite right.

11.  Not sure why the title is “Erastin Rapidly inhibits…”? Figs.1E,F show it takes 3 hours to significantly inhibit P-gp activity.